# SMOOTHED-SGDMAX: A STABILITY-INSPIRED ALGORITHM TO IMPROVE ADVERSARIAL GENERALIZATION

## ABSTRACT

Unlike standard training, deep neural networks can suffer from serious overfitting problems in adversarial settings. Recent research (Xing et al., 2021b; Xiao et al., 2022) suggested that adversarial training can have nonvanishing generalization error even if the sample size $n$ goes to infinity. A natural question arises: can we eliminate the generalization error floor in adversarial training? This paper gives an affirmative answer. First, by an adaptation of information-theoretical lower bound on the complexity of solving Lipschitz-convex problems using randomized algorithms, we establish a minimax lower bound $\Omega(s(T)/n)$ given a training loss of $1/s(T)$ for the generalization gap in non-smooth settings, where $T$ is the number of iterations, and $s(T) \to +\infty$ as $T \to +\infty$. Next, by observing that the nonvanishing generalization error of existing adversarial training algorithms comes from the non-smoothness of the adversarial loss function, we employ a smoothing technique to smooth the adversarial loss function. Based on the smoothed loss function, we prove that a smoothed version of SGDmax algorithm can achieve a generalization bound $\mathcal{O}(s(T)/n)$, which eliminates the generalization error floor and matches the minimax lower bound. Experimentally, we show that the Smoothed-SGDmax algorithm improves adversarial generalization on common datasets.

## 1 INTRODUCTION

Deep neural networks (DNNs) (Krizhevsky et al., 2012; Hochreiter & Schmidhuber, 1997) is successful and rarely suffered overfitting issues (Zhang et al., 2021). This phenomenon is also called benign overfitting. A well-trained neural network model can generalize well to the test data. However, in adversarial machine learning, overfitting becomes a serious issue (Rice et al., 2020). Before the training algorithms converge, the robust test error starts to increase. This special type of overfitting is called *robust overfitting* and can be observed in the experiments on common datasets. See Fig. 1, orange curve. Therefore, mitigating the robust overfitting is important to increase the adversarial robustness of a DNN model. Several recent works tried to figure out the causes of robust overfitting and designed methods to mitigate it. See the discussion in Sec. 2.

A recent line of work (Xing et al., 2021b; Xiao et al., 2022) studied the robust overfitting issue of adversarial training from a theoretical perspective, using the notion of uniform algorithmic stability. Uniform algorithmic stability (UAS) (Bousquet & Elisseeff, 2002) was introduced to bound the generalization gap in machine learning problems. It provides algorithm-specific generalization bounds instead of algorithm-free generalization bounds such as classical results on VC-dimension (Vapnik & Chervonenkis, 2015) and Rademacher complexity (Bartlett & Mendelson, 2002). Such stability-based generalization bounds provide insight into understanding the generalization ability of neural network models trained by different algorithms.

Traditional adversarial training is to perform stochastic gradient descent (SGD) on the max function of the standard counterpart, which is also called SGDmax (Farnia & Ozdaglar, 2021). We will not distinguish two algorithms, "SGDmax" and "adversarial training (AT)", in the paper. The work of (Xing et al., 2021b; Xiao et al., 2022) both showed that SGDmax incurs a stability-based generalization bound in $\mathcal{O}(c(T) + s(T)/n)$. Here $T$ is the number of iterations, $n$ is the number of samples, $s(T)$ is a function satisfies $s(T) \to +\infty$ as $T \to +\infty$, and $c(T)$ is a sample size-independent

Table 1: Comparison of stability-based generalization bounds of adversarial generalization gap. $c_1(T)$ and $c_2(T)$ are sample size-independent terms. Details of the form of $s(T)$, $c_1(T)$, $c_2(T)$ are discussed in Sec. 4 and Sec. 5.

|  | Upper Bounds | Worst-case Lower Bounds | Achieves minimax lower bound $\Omega(s(T)/n)$ |
|---|---|---|---|
| SGDmax | $\mathcal{O}(c_1(T) + \frac{s(T)}{n})$ | $\Omega(c_2(T) + \frac{s(T)}{n})$ | ✗ |
| **Smoothed-SGDmax** | $\mathcal{O}(\frac{s(T)}{n})$ | $\Omega(\frac{s(T)}{n})$ | ✓ |

term and increase with $T$. Details of the form of $s(T)$, $c(T)$ are discussed in Sec. 4 and Sec. 5. They also provided the matching lower bounds to show that the sample size-independent term is unavoidable for SGDmax-based adversarial training algorithms. It provides a possible explanation of robust overfitting: even though we have arbitrarily large number of training samples, the adversarial generalization gap still does not vanish. The first question arises: what is the lower bound of the generalization gap for algorithms in adversarial machine learning settings? To answer this question, we develop a minimax lower bound, $\Omega(s(T)/n)$, for the generalization gap in non-smoothing settings when the training loss is $1/s(T)$. Clearly, SGDmax does not achieve the lower bound. Therefore, we are motivated to design algorithms to reduce the non-vanishing sample size-independent term. The following main question of our paper arises:

*Can we eliminate the error floor in generalization bounds of adversarial generalization gap?*

We call the term $c(T)$ as generalization error floor. It is observed that the term $c(T)$ comes from the non-smoothness of the adversarial loss. Hence, stability analysis on some smoothed algorithms has been studied recently. It includes noise-SGD and differential privacy-SGD (Bassily et al., 2020), adding noise to weight and data (Xing et al., 2021b), stochastic weight averaging, and cyclic learning rate (Xiao et al., 2022). Unfortunately, these smoothed algorithms cannot eliminate the generalization error floor.

In this paper, we employ a smoothing technique using tools from Moreau envelope function to smooth the adversarial loss and perform gradient descent to this smooth surrogate. Following the name SGDmax, we refer the smoothed version of SGDmax as Smoothed-SGDmax, which improves adversarial generalization. We prove that Smoothed-SGDmax has the same training loss $1/s(T)$ on adversarial loss. Most importantly, Smoothed-SGDmax eliminates the generalization error floor and achieves the minimax lower bound $\Omega(s(T)/n)$ of the generalization gap. The comparison of the stability-based generalization upper bound and lower bound of our proposed algorithm with the SGDmax-based adversarial training algorithm is given in Table 1. Additionally, our proposed algorithm

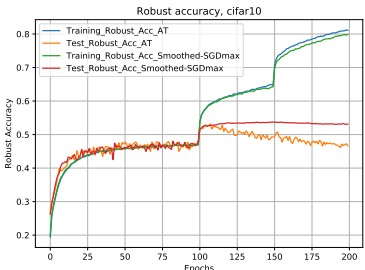

Figure 1: Experiments of adversarial training and Smoothed-SGDmax on CIFAR-10.

can be viewed as a general form of stochastic weight averaging (SWA (Izmailov et al., 2018)). As a by-product, we provide an understanding of SWA in our framework, see more discussion in Sec. 5.4. In Fig. 1, we show the training procedure of our proposed algorithm as well as adversarial training on CIFAR-10.

The contributions of our work are listed as follows:

1. Main result: we prove that the generalization error floor in non-smooth loss minimization can be eliminated by a properly designed algorithm, which we called Smoothed-SGDmax.

2. We develop the minimax lower bound of the generalization gap in non-smooth loss minimization. Specifically, we show that an algorithm has at least $\Omega(s(T)/n)$ generalization gap if the training error training loss is $1/s(T)$. Smoothed-SGDmax achieves the minimax lower bound.

3. Experiments on common datasets verify the theoretical results and show the effectiveness of our proposed algorithm in practice.

## 2 RELATED WORK

**Adversarial Robustness.** Starting from the work of (Szegedy et al., 2013), it has now been well known that deep neural networks trained via standard gradient descent based algorithms are highly susceptible to imperceptible corruptions to the input data (Goodfellow et al., 2014; Chen et al., 2017; Carlini & Wagner, 2017). Adversarial training and its variants are proposed to improve the adversarial robustness of DNNs (Madry et al., 2017; Wu et al., 2020; Gowal et al., 2020). *Robust Overfitting.* Starting from the work of (Rice et al., 2020), a series of work studied the causes of robust overfitting. (Yu et al., 2022) studied robust overfitting from the perspective of adversarial distribution. (Chen et al., 2021) leveraged knowledge distillation and self-training to mitigate robust overfitting.

**Learning Theory for Adversarial Generalization.** *Classical learning theory.* The work of (Attias et al., 2021; Montasser et al., 2019) explained generalization in adversarial settings using VC-dimension. The work of (Yin et al., 2019; Khim & Loh, 2018) studied the poor generalization of adversarial training using tools from Rademacher complexity. However, VC-dimension and Rademacher complexity are algorithm-independent bounds for generalization. They cannot reveal the effect of algorithms on generalization. *Other theoretical analysis.* (Sinha et al., 2017) study the generalization of an adversarial training algorithm in terms of distributional robustness. The work of (Xing et al., 2021a;c; Javanmard et al., 2020) studied the generalization properties in the setting of linear regression. Gaussian mixture models are used to analyze adversarial generalization (Taheri et al., 2020; Javanmard et al., 2020; Dan et al., 2020). The work of (Allen-Zhu & Li, 2020) explains adversarial generalization through the lens of feature purification.

**Uniform Stability.** Stability can be traced back to the work of (Rogers & Wagner, 1978). In statistical learning problems, it was well developed in analyzing the algorithm-based generalization bounds (Bousquet & Elisseeff, 2002). These bounds have been significantly improved in a recent sequence of works (Feldman & Vondrak, 2018; 2019). The work of (Chen et al., 2018) discussed the optimal trade-off between stability and convergence. (Bassily et al., 2020) studied the stability of SGD on non-smooth loss. They proved that the generalization bound contains a sample size-independent term. The work of (Xing et al., 2021b; Xiao et al., 2022) showed that adversarial loss is non-smooth and SGDmax-based adversarial training algorithms will incur the generalization error floor.

## 3 PRELIMINARIES: STABILITY ANALYSIS FOR GENERALIZATION GAP

Let $\mathcal{D}$ be an unknown distribution in the sample space $\mathcal{Z}$. Let $S = \{z_1, \ldots, z_n\} \sim \mathcal{D}^n$ be an sample dataset drawn i.i.d. according to $\mathcal{D}$. Our goal is to find a model $w$ with small population risk, defined as:

$$R_{\mathcal{D}}(w) = \mathbb{E}_{z \sim \mathcal{D}} h(w, z),$$

where $h(\cdot, \cdot)$ is the loss function. Since we cannot minimize the objective $R_{\mathcal{D}}(w)$ directly, we instead minimize the empirical risk, defined as

$$R_S(w) = \frac{1}{n} \sum_{i=1}^{n} h(w, z_i).$$

Let $\bar{w}$ be the optimal solution of $R_S(w)$. Then, for the algorithm output $\hat{w} = A(S)$, we define the expected generalization gap as

$$\mathcal{E}_{gen}(A, h, n, \mathcal{D}) = \mathbb{E}_{S \sim \mathcal{D}^n, A}[R_{\mathcal{D}}(A(S)) - R_S(A(S))]. \tag{3.1}$$

We define the the expected optimization gap as

$$\mathcal{E}_{opt}(A, h, n, \mathcal{D}) = \mathbb{E}_{S \sim \mathcal{D}^n, A}[R_S(A(S)) - R_S(\bar{w})]. \tag{3.2}$$

We use $\mathcal{E}_{gen}$ and $\mathcal{E}_{opt}$ as short hand notations of the above definition. To bound the generalization gap of a model $\hat{w} = A(S)$ trained by a randomized algorithm $A$, we employ the following notion of *uniform stability*.

**Definition 3.1.** *A randomized algorithm $A$ is $\varepsilon$-uniformly stable if for all data sets $S, S' \in \mathcal{Z}^n$ such that $S$ and $S'$ differ in at most one example, we have*

$$\sup_z \mathbb{E}_A \left[ h(A(S); z) - h(A(S'); z) \right] \leq \varepsilon.$$ (3.3)

The following theorem shows that expected generalization gap can be attained from uniform stability.

**Theorem 3.1** (Generalization in expectation (Hardt et al., 2016)). *Let $A$ be $\varepsilon$-uniformly stable. Then, the expected generalization gap satisfies*

$$|\mathcal{E}_{gen}| = |\mathbb{E}_{S,A}[R_{\mathcal{D}}[A(S)] - R_S[A(S)]]| \leq \varepsilon.$$

**Uniform Argument Stability (UAS).** If $h$ is $L$-Lipschitz, *i.e.,* $|h(w_1; z) - h(w_2; z)| \leq L\|w_1 - w_2\|$, we can use UAS$= \mathbb{E}\|A(S) - A(S)'\|$ to measure the generalization gap.

## 3.1 SGDMAX INCURS GENERALIZATION ERROR FLOOR

**Adversarial Loss.** In adversarial training, we consider the following adversarial loss

$$h(w; z) = \max_{\|z - z'\| \leq \epsilon} g(w; z'),$$ (3.4)

where $g(w; z)$ is the loss function of the standard counterpart. In practice, $w$ is usually the parameter of neural networks.

**Generalization Error Floor.** As discussed in (Xing et al., 2021b; Xiao et al., 2022), even if $g$ is a smooth function, $h$ is not necessarily smooth. They assumed $h$ to be generally non-smooth or $\eta$-approximately smooth, which is a subset of non-smooth functions. Under both assumptions, there exist non-vanishing terms in the bounds of UAS:

$$c_1(T) + \frac{LT\alpha}{n} \leq \text{UAS} \leq c_2(T) + \frac{LT\alpha}{n},$$ (3.5)

where the forms of $c_1(T)$ and $c_2(T)$ are listed in Table 2. We refer $c_1(T)$ and $c_2(T)$ as generalization error floors.

Table 2: Generalization error floor in previous studies.

|  | Assumption on $h$ | Upper Bounds $c_1(T)$ | Lower Bounds $c_2(T)$ |
|---|---|---|---|
| (Xing et al., 2021b) | non-smooth | $\mathcal{O}(L\alpha\sqrt{T})$ (Prop. 1) | $\Omega(\alpha\sqrt{T})$ (Thm. 1) |
| (Xiao et al., 2022) | $\eta$-approx-smooth | $\mathcal{O}(\eta\alpha T)$ (Thm 5.1) | $\Omega(\eta\alpha\sqrt{T})$ (Thm. 5.2) |

## 4 MINIMAX LOWER BOUND

Following the work of (Xing et al., 2021b), we mainly consider the following function class of convex, non-smooth, and Lipschitz functions throughout the paper.

$$\mathcal{H} = \{ h : W \times \mathcal{Z} \to \mathbb{R} \mid h \text{ is convex, } L\text{-Lipschitz in } w, |W| = D_W \}.$$ (4.1)

$L$-Lipschitz is a standard assumption in uniform stability analysis since (Hardt et al., 2016). The assumption of convexity is to compare with the existing results and to develop the following the minimax lower bound.

**Definition 4.1** (Training Loss). *We say an algorithm class $\mathcal{A}$ has training loss $1/s(T)$ on a function class $\mathcal{H}$, if for all $A \in \mathcal{A}$ and $h \in \mathcal{H}$, running $A$ on $h$ for $T$ iterations, we have*

$$\mathcal{E}_{opt}(A, h, n, \mathcal{D}) \leq \mathcal{O}\left(\frac{1}{s(T)}\right),$$

*where $\lim_{T \to +\infty} s(T) = +\infty$.*

**Proposition 4.1** (Minimax lower bound of generalization gap). *Let $\mathcal{H}$ be the function class defined in Eq. (4.1). Let $\mathcal{A}$ be the class of randomized algorithms using $n$ samples with training loss $1/s(T)$ on $\mathcal{H}$. For all $n$, there exists $T$, s.t. the following lower bound holds.*

$$\min_{A \in \mathcal{A}} \max_{\mathcal{D}} \mathcal{E}_{gen}(A, h, n, \mathcal{D}) \geq \Omega\left(\frac{s(T)}{n}\right). \tag{4.2}$$

The proof of Prop. 4.1 is based on a lower bound of the complexity of Lipschitz-convex problems ((Nemirovskij & Yudin, 1983), Ch.4), see Appendix A.1.

Clearly, SGDmax can not achieve the minimax lower bound.

## 5    SMOOTHED-SGDMAX: ELIMINATING GENERALIZATION ERROR FLOOR

In this section, we will design an algorithm satisfying the following two properties:

1. It has the same training loss as the SGDmax algorithm;

2. Suppose it achieves $1/s(T)$ training loss after $T$ iterations. Then, the generalization bound is bounded by $s(T)/n$.

### 5.1    SMOOTH SURROGATE ADVERSARIAL LOSS

The non-smoothness of $h$ leads to a poor generalization bound. This motivates us to construct smooth surrogate loss functions to improve adversarial generalization. Inspired by the work of (Zhang & Luo, 2020), we use the Moreau envelope function to smooth the adversarial loss. Let

$$K(w, u; z) = h(w; z) + \frac{p}{2}\|w - u\|^2. \tag{5.1}$$

If $h$ is $l$-weakly convex, we can choose $p > l$ to insure that $K(w, u; z)$ is strongly convex with respect to $w$. In the case that $h$ is convex, we only need $p > 0$. We define the Moreau envelope function:

$$M(u; S) = \min_{w \in W} K(w, u; S) = \min_{w \in W} \frac{1}{n}\sum_{z \in S} K(w, u; z), \tag{5.2}$$

$$w(u; S) = \arg\min_{w \in W} K(w, u; S). \tag{5.3}$$

Then, $M(u; S)$ is a smooth function. Formally, we state the theoretical results as follows.

**Lemma 5.1.** *Assume that $h$ is $l$-weakly convex. Let $p > l$. Then, $M(u; S)$ satisfies*

1. $\min_u M(u; S)$ *has the same global solutions as* $\min_w R_S(w)$.

2. *The gradient of $M(u; S)$ is $\nabla_u M(u; S) = p(u - w(u; S))$.*

3. *$M(u; S)$ is $pl/(p - l)$-weakly convex.*

4. *$M(u; S)$ is $(2p^2 - pl)/(p - l)$-gradient Lipschitz continuous.*

5. *$M(u; S)$ has bounded gradient norm $L$.*

**Remark:**    The proof of Lemma 5.1 is due to (Rockafellar, 1976) and also provided in Appendix A.1. We focus on the case where $h$ is convex in the main text. Then, Lemma 5.1.3 and 5.1.4 reduce to $M(u; S)$ is convex and $2p$-gradient Lipschitz. Lemma 5.1 is stated in general $l$-weakly convex cases for further theoretical studies. Since $M(u; S)$ has the same global solutions as $R_S(w)$, we can do adversarial training using this smooth objective $M(u; S)$. A natural way is to perform gradient descent to $M(u; S)$. By Lemma 5.1, the estimate of the gradient requires the estimate of the solution of the minimization problem $\min_w K(w, u; S)$. Depending on whether we solve the subproblems exactly or not, we have the exact approach and inexact approach.

## 5.2 EXACT APPROACH

We first consider the exact approach, which is the gradient descent to $M(u; S)$.

**Theorem 5.1.** *Assume $h$ is a convex, L-Lipschitz function. Suppose we run GD on the smoothed surrogate adversarial loss $M(u; S)$ defined in Eq. (5.2) with fixed stepsize $\alpha \leq 1/\sqrt{T}$ for $T \geq 4p^2$ steps. Then, the optimization and generalization gap satisfies*

$$\mathcal{E}_{opt} \leq \mathcal{O}(1/T\alpha) \quad and \quad \mathcal{E}_{gen} \leq \left(\frac{2L^2 T\alpha}{n}\right). \tag{5.4}$$

**Remark:** Thm. 5.1 is not obtained from the work of (Hardt et al., 2016). Notice that

$$M(u; S) = \min_{w \in W} \frac{1}{n} \sum_{z \in S} K(w, u; z) \neq \frac{1}{n} \sum_{z \in S} \min_{w \in W} K(w, u; z).$$

$\min_u M(u; S)$ is not a finite sum problem. However, the analysis in (Hardt et al., 2016) can only be applied to finite sum problems. Thm. 5.1 requires a different proof. In summary, there are two steps: 1) Build the recursion from $\|u_S^t - u_{S'}^t\|$ to $\|u_S^{t+1} - u_{S'}^{t+1}\|$; 2) Unwind the recursion. The main challenge comes from the first step. To this end, we develop a new error bound and a different decomposition to build the recursion. Details are deferred to Appendix A.3. Thm. 5.1 is our first main result. It shows that the exact approach achieves the minimax lower bounds of the generalization gap. The extension to weakly-convex cases is provided in Appendix B.

However, the exact approach requires the exact minimization of $K(w, u; S)$, which is sometimes computationally intractable. To address this issue, we consider the inexact approach below.

## 5.3 THE INEXACT APPROACH

The inexact approach is to estimate $\nabla_u M(u; S)$ by inexactly solving $\min_w K(w, u; S)$. To this aim, we perform multiple steps of SGD to the subproblem $\min_w K(w, u; S)$, attaining an estimate $\bar{w}(u)$ of the true $w(u)$, and then use $\bar{w}(u)$ to estimate $\nabla_u M(u; S)$.

---

**Algorithm 1** Smoothed-SGDMax

1: Initialize $w^0$, $u^0$;
2: Choose stepsize $c_s^t > 0$ and $\alpha_t > 0$;
3: **for** $t = 0, 1, 2, \ldots, T$ **do**
4:     Let $w_0^t = w^t$;
5:     **for** $s = 0, 1, 2, \cdots, N$ **do**
6:         Draw a sample $z_s^t$ from $S$ uniformly;
7:         $w_{s+1}^t = P_W(w_s^t - c_s^t \nabla_w K(w_s^t, u^t; z_s^t))$;
8:     **end for**
9:     $w^{t+1} = w_N^t$;
10:    $u^{t+1} = u^t + \alpha_t p(w^{t+1} - u^t)$;
11: **end for**

---

In Step 7 in Alg. 1, we run SGD on $K(w, u, S)$ *w.r.t* $w$ to find a solution given $u$. In step 10, we run GD on $K(w, u, S)$ *w.r.t* $u$. To provide the upper bounds of the optimization gap and generalization gap of Alg. 1, we need the following Lemma for the inner optimization.

**Lemma 5.1.** *Given $t$ and $u^t$, suppose we run SGD on $K(w, u^t, S)$ w.r.t. $w$ with stepsize $c_s^t \leq 1/(p-l)s$ for $N$ steps. $w_N^t$ is approximately the minimizer with an error $C_1^2/N$, i.e.,*

$$E\|w_N^t - w(u^t)\|^2 \leq \frac{C_1^2}{N},$$

*where $C_1 = (L + pD_W)/(p - l)$.*

In convex case, *i.e.*, $l = 0$, we have $C_1 = L/p + D_W$. Lemma 5.1 provides the optimization error of the inner loop. In words, if we run the inner loop for sufficient steps, we can approximate the smoothed loss $M(u; S)$. Below we provide the training loss and uniform stability of Smoothed-SGDmax with sufficient steps for the inner loop.

**Theorem 5.2** (training loss of Smoothed-SGDmax). *Suppose $h$ is convex and $L$-Lipschitz. In Alg. 1, if we choose inner stepsize $c_s^t \leq 1/ps$, number of steps in inner loop $N = T$, outer stepsize $\alpha \leq 1/\sqrt{T}$, $T \geq 4p^2$, the optimization gap satisfies*

$$\mathcal{E}_{opt} \leq \frac{\|u^0 - u^*\|^2 + 2pC_1 D_W + (L + pD_W)^2}{2T\alpha} = \frac{C_2}{T\alpha}, \tag{5.5}$$

*where $C_2 = \|u^0 - u^*\|^2/2 + pC_1 D_W + (L + pD_W)^2/2$.*

**Theorem 5.3** (Generalization bound of Smoothed-SGDmax). *Assume that $h$ is convex and $L$-Lipschitz. In Alg. 1, if we choose inner stepsize $c_s^t \leq 1/ps$, number of steps in inner loop $N = n^2$, outer stepsize $\alpha_t \leq 1/\sqrt{T}$, $T \geq 4p^2$, the generalization gap satisfies*

$$\mathcal{E}_{gen} \leq L\left(\frac{2C_1 p}{n} + \frac{2L}{n}\right)\sum_{t=1}^{T}\alpha_t = \frac{C_3}{n}\sum_{t=1}^{T}\alpha_t, \tag{5.6}$$

*where $C_3 = L(4L + 2pD_W)$.*

Thm. 5.2 and 5.3 are the main results of our paper. For fixed stepsize $\alpha_t = \alpha$, it shows that Alg. 1 has training loss $\mathcal{O}(1/T\alpha)$ and has optimal generalization bound in $\mathcal{O}(T\alpha/n)$.

**Interpretation of Number of Steps.** In practice, if we use batch size 1 and go through the whole dataset in each epoch, $T$ can be viewed as the number of epochs, and $N$ can be viewed as the number of samples. Let $T\alpha = \sqrt{C_2 n/C_3}$, we obtain the optimal excess risk with respect to $T$ and $\alpha$, i.e., $\mathcal{E}_{opt} + \mathcal{E}_{gen} \leq 2\sqrt{\frac{C_2 C_3}{n}}$.

## 5.4 FURTHER COMPARISON WITH EXISTING ALGORITHMS

In Alg. 1, Step 7 is just to run SGD on $K(w, u; z) = h(w; z) + p\|w - u\|^2/2$ instead of $h(w; z)$. The additional term can be viewed as a regularization term similar to weight decay. Step 10 is a model averaging step similar to stochastic weight averaging (SWA). We compare Smoothed-SGDmax with some existing algorithms in detail. The summary of the comparison is provided in Table 3. We can see that only Smoothed-SGDmax can reduce the generalization error floor.

Table 3: Comparison of SGDmax, weight decay, proximal update, stochastic weight averaging, and Smoothed-SGDmax. Only Smoothed-SGDmax reduces the error floor in the generalization bound.

|  | Operation on $w$ | Operation on $u$ | No error floor |
|---|---|---|---|
| SGDmax | Minimize $w$ on $R_S(w)$ | No operation on $u$ | ✗ |
| Weight decay | Minimize $w$ on $K(w, u; S)$ | Set $u = 0$ | ✗ |
| Proximal update | Minimize $w$ on $R_S(w)$ | Update rule in Eq. (5.8) | ✗ |
| SWA | Minimize $w$ on $R_S(w)$ | Minimize $u$ on $K(w, u; S)$ | ✗ |
| **Smoothed-SGDmax** | Minimize $w$ on $K(w, u; S)$ | Minimize $u$ on $K(w, u; S)$ | ✓ |

**Weight Decay.** Weight decay (WD) is to add a $\ell_2$ regularization to the empirical loss. The loss function with WD is $h(w; z) + p\|w\|^2/2$. Therefore, if we replace Step 10 by $u = 0$ in Alg. 1, Smoothed-SGDmax reduces to a simple weight decay regularization technique. Following the analysis in Table 2, it is easy to see that adversarial training with weight decay incurs a generalization bound in

$$\mathcal{E}_{gen} \leq 2L(L_z \epsilon + L/n)T\alpha, \tag{5.7}$$

where the step size $\alpha \leq 1/(L_w - p)$. Therefore, weight decay is not guaranteed to reduce the additional sample size-independent term.

**Proximal Update.** The proximal update is to apply an update rule,

$$P_{f,\alpha}(w) = \arg\min_{u} \frac{1}{2}\|w - u\|^2 + \alpha f(u), \tag{5.8}$$

after a stochastic gradient update. Both proximal update and Smoothed-SGDmax use the Moreau envelope function, but the algorithms are different. The stability analysis of the proximal update is given in (Hardt et al., 2016), Def. 4.5 and Lemma 4.6. It is proved that the proximal update is 1-expansive if $f$ is convex. Therefore, the generalization bound of the proximal update is no larger than that of SGD. In non-smooth cases, SGD incurs an error floor. The proximal update is not guaranteed to eliminate the error floor.

**Stochastic Weight Averaging.** Stochastic weight averaging suggests using the weighted average of the iterates rather than the final one for inference. The update rules of SWA is $u^{t+1} = \tau^t u^t + (1 - \tau^t)w^{t+1}$. In the work of (Xiao et al., 2022), they provide a generalization bound for SWA in the case that $u$ is the average of the iterates, which is equivalent to using the step size $u^t = (t-1)/t$. The generalization bound in this case is

$$\mathcal{E}_{gen}(SWA) \leq (LL_z\epsilon + 2L^2/n)T\alpha. \tag{5.9}$$

The sample size-independent term is one-half of the one without SWA. However, the additional term is still unavoidable in the analysis. SWA is still not guaranteed to achieve the minimax lower bound in this analysis.

**Optimal Generalization Bound of SWA in our Regime.** In Alg. 1, if we denote $\tau^t = 1 - \alpha^t p$, Step 10 can be view as a weight averaging step. In Thm. 5.3, it is required that $\alpha^t \leq 1/2p$. Then, $\tau^t = (1 - \alpha^t p) \geq 1/2$. Therefore, by fixing $\alpha^t p$ to be constant and letting $p \to 0$, our proposed algorithm is reduced to SWA. In other words, our proposed algorithm can be viewed as a general form of SWA. Also, we provide an optimal generalization bound of SWA in the regime that $\tau \in [1/2, 1]$ and $p \to 0$.

## 6 EXPERIMENTS

**Training Procedure of Smoothed-SGDmax.** To have a first glance of how Smoothed-SGDmax mitigates robust overfitting, we consider the experiments on a lightweight model, PreActResNet-18, on CIFAR-10, CIFAR-100, and SVHN to plot the training procedure.

**Training Settings.** For the attack algorithms, we use $\ell_\infty$-PGD-10 (Madry et al., 2017), $\epsilon = 8/255$. The step size is set to be $\epsilon/4$. For adversarial training, we use piece-wise learning rates, which are equal to $0.1, 0.01, 0.001$ for epochs 1 to 100, 101 to 150, and 151 to 200, respectively. For Smoothed-SGDmax, we keep the piece-wise learning rate (for the choice of $c_s^t$ in Alg. 1) for comparison. Because of the similarity of $\ell_2$ regularization term of weight decay and the proximal term in $K(w, u; z)$, we set $p = 5 \times 10^{-4}$, which is a common choice of weight decay. The step size $\alpha_t$ of updating $u$ is set to be 50, then $\tau = 1 - \alpha p = 0.995$.

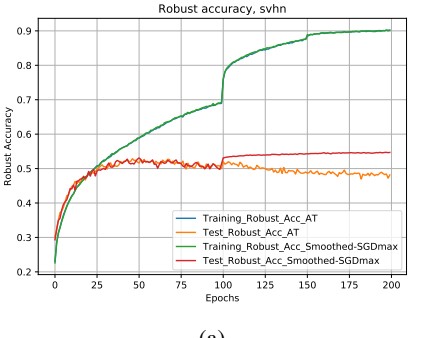
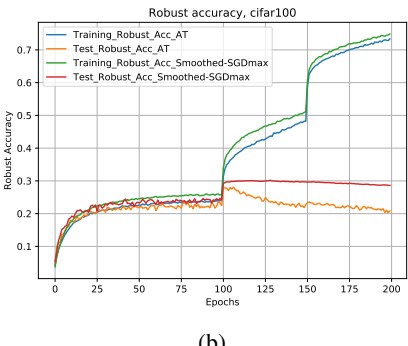

|     |     |
|:---:|:---:|
| (a) | (b) |

Figure 2: Robust test accuracy of adversarial training and Smoothed-SGDmax on SVHN and CFAR-100.

The training procedure of the experiments on CIFAR-10 is already provided in Introduction, Fig. 1. The experiments on SVHN and CIFAR-100 are provided in Fig. 2. For adversarial training, the robust test accuracy starts to decrease at around the $100^{th}$ epoch, which is called robust overfitting (Rice et al., 2020). Using Smooth-SGDmax, the robust overfitting issue is much milder. These experiments verify the generalization bounds. The bound of Smoothed-SGDmax (which is $\mathcal{O}(T\alpha/n)$) is much better than the bound of adversarial training ($\mathcal{O}(T\alpha + T\alpha/n)$).

**Sample Complexity.** Secondly, we study the sample complexity provided in Thm. 5.3. We use Wide-ResNet-28 $\times$ 10 with Swish activation function for better test accuracy instead of ResNet-18. The training setting mainly follows the work of (Gowal et al., 2020). We consider two losses, adversarial loss (Madry et al., 2017) and TRADES loss (Zhang et al., 2020) for the choice of $h(w; z)$. The total number of epochs is 400. Other training settings are similar to the experiments on ResNet-18.

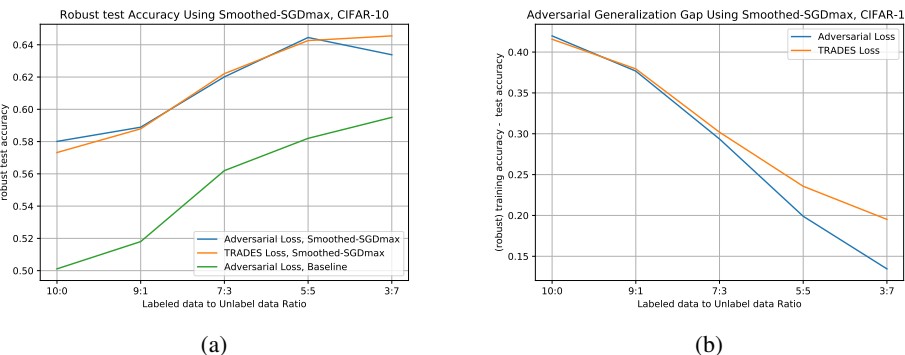

(a)                                          (b)

Figure 3: Robust test accuracy and generalization gap in the experiments of training CIFAR-10 using Smoothed-SGDmax.

**Adversarial Generalization Gap.** CIFAR-10 only contains 50K training samples. We adopt the pseudo-label data introduced in (Carmon et al., 2019) to study the sample complexity. Increasing the percentage of pseudo-label data is an approximation of increasing the training data. In Fig. 3, we show the robust test accuracy (a) and adversarial generalization gap (b). The results are consistent with the theorem that Smoothed-SGDmax reduces a term in the generalization bounds.

Table 4: Robust test accuracy of our proposed algorithm. $\epsilon = 8/255$. Model: WideResNet-28 $\times$ 10 with Swish activation function. Training data: Labeled to unlabeled data ratio: 3:7.

| Dataset | Loss | Algorithm | Clean | AutoAttack |
|---|---|---|---|---|
| CIFAR-10 | AT Loss | SGDmax | 90.93±0.25% | 58.41±0.25% |
| | | Smooth-SGDmax | 91.51±0.20% | 59.14±0.18% |
| | TRADES Loss | SGDmax | 88.36% | 59.45% |
| | | Smooth-SGDmax | 85.33±0.13% | 62.41±0.11% |
| CIFAR-100 | TRADES Loss | SGDmax | 59.38% | 26.07% |
| | | Smooth-SGDmax | 59.25±0.22% | 28.54±0.19% |

In Table 4, we provide the robust test performance of our proposed algorithms. The baseline performance on CIFAR-10 are reported in (Gowal et al., 2020). We can see that the performance of our proposed algorithms is comparable in the same settings used in (Gowal et al., 2020). Notice that the state-of-the-art performance of adversarial robustness is obtained using large models (*e.g.,* WideResNet-106 $\times$ 16) and DDPM-generated data (Rebuffi et al., 2021). We do not have enough resources to run large models.

## 7 CONCLUSION

In this paper, we study a question: can we design an algorithm to eliminate the generalization error floor of the adversarial generalization gap? By using tools from Moreau envelopes, we consider a smoothed version of SGDmax. We prove that it has the same convergence guarantee as SGDmax and attains the minimax lower bound of the generalization gap in non-smooth loss minimization. Most importantly, Smoothed-SGDmax can eliminate the generalization error floor. We hope our work can lead to a better understanding of adversarial machine learning theory.

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

# A  PROOF OF THEOREMS

## A.1  PROOF OF PROPOSITION 4.1

The proof is adopted from the proof of minimax lower bound of optimization error from the work of (Chen et al., 2018). We define the excess risk as $R_\mathcal{D}(w) - \min_{w \in W} R_\mathcal{D}(w)$. A minimax lower bound of the excess risk for the function class $\mathcal{H}$ is given in (Nemirovskij & Yudin, 1983):

$$\min_w \max_\mathcal{D} \mathbb{E}_{S \sim \mathcal{D}^n}[R_\mathcal{D}(w) - \min_{w \in W} R_\mathcal{D}(w)] \geq \frac{LD_W}{C_4 \sqrt{n}}, \tag{A.1}$$

where $C_4$ is a universal constant. By the excess risk decomposition, we have

$$\mathbb{E}_{S \sim \mathcal{D}^n}[R_\mathcal{D}(w) - \min_{w \in W} R_\mathcal{D}(w)] \leq \mathcal{E}_{opt}(w) + \mathcal{E}_{opt}(w). \tag{A.2}$$

Let $A \in \mathcal{A}$ and $w^T$ be the algorithm output of $A$, we have $\mathcal{E}_{opt}(w^T) \leq \mathcal{O}(1/s(T))$. Then,

$$\min_{A \in \mathcal{A}} \max_\mathcal{D} \mathcal{E}_{gen}(w^T) \geq \Omega \left( \frac{LD_W}{\sqrt{n}} - \frac{1}{s(T)} \right). \tag{A.3}$$

Complete the square, we have

$$\frac{LD_W}{\sqrt{n}} - \frac{1}{s(T)} = - \left( \frac{1}{\sqrt{s(T)}} - \frac{LD_W \sqrt{s(T)}}{2n} \right)^2 + \frac{L^2 D_W^2 s(T)}{4n}. \tag{A.4}$$

Since $s(T) \to +\infty$ as $n \to +\infty$, we can choose $T$ s.t. $\frac{1}{\sqrt{s(T)}}$ is close to $\frac{LD_W \sqrt{s(T)}}{2n}$. Therefore, there exists $T$, s.t.

$$\min_{A \in \mathcal{A}} \max_\mathcal{D} \mathcal{E}_{gen} \geq \Omega \left( \frac{s(T)}{n} \right). \tag{A.5}$$

$\square$

## A.2  PROOF OF LEMMA 5.1

To simplify the notation, we use $M(u)$ as a short hand notation of $M(u; S)$. Similar to $h(u)$, $K(u)$, and $w(u)$.

1. Let $w^* \in \arg\min R_S(w)$. We have

$$R_S(w^*) = K(w^*, u = w^*, S) \geq K(w(u), u = w^*, S) \geq R_S(w(u = w^*)).$$

Then, the equality holds. Therefore, $w = u = w^*$ is the optimal solution of both $\min_w R_S(w)$ and $\min_u M(u; S)$.

2. Since $K(w, u)$ is a $(p-l)$-strongly convex function, $w(u)$ is unique. Then

$$M(u) = h(w(u)) + \frac{p}{2} \|w(u) - u\|^2.$$

By taking the derivative of $M(u)$ with respect to $u$, we have

$$\nabla_u M(u) = \left[ \frac{\partial w(u)}{\partial u} \right]^T \cdot \nabla_{w(u)} h(w(u)) + \left[ \frac{\partial w(u)}{\partial u} - I \right]^T \cdot p(w(u) - u). \tag{A.6}$$

$$= \left[ \frac{\partial w(u)}{\partial u} \right]^T \cdot (\nabla_{w(u)} h(w(u)) + p(w(u) - u)) + p(u - w(u)). \tag{A.7}$$

Since $w(u)$ is the optimal solution of $K(w, u)$, we have

$$\nabla_{w(u)} K(w(u), u) = \nabla_{w(u)} h(w(u)) + p(w(u) - u) = 0. \tag{A.8}$$

Therefore, the first term in A.7 is equal to zero. We have $\nabla_u M(u) = p(u - w(u))$.

3. In Eq. (A.8), take the derivatives with respect to $u$ on both sides, we have

$$\left[\frac{\partial w(u)}{\partial u}\right]^T \nabla_w^2 h(w) + p(\left[\frac{\partial w(u)}{\partial u}\right]^T - I) = 0. \tag{A.9}$$

Organizing the terms, we have

$$\left[\frac{\partial w(u)}{\partial u}\right]^T (\nabla_w^2 h(w) + pI) = pI. \tag{A.10}$$

Since $h(w)$ is $l$-weakly convex, $\nabla_w^2 h(w) + pI$ is positive definite. Then,

$$\left[\frac{\partial w(u)}{\partial u}\right]^T \prec \frac{p}{p-l} I. \tag{A.11}$$

Then,

$$\nabla_u^2 M(u) = [\frac{\partial}{\partial u} p(u - w(u))]^T = p(I - \left[\frac{\partial w(u)}{\partial u}\right]^T) \succ p(1 - \frac{p}{p-l}) I. \tag{A.12}$$

Therefore, $M(u)$ is a $pl/(p-l)$-weakly convex function.

4. By Eq. (A.11), we have

$$\|\nabla M(u_1) - \nabla M(u_2)\| = p\|u_1 - w(u_1) - u_2 - w(u_2)\| \le p(1 + \frac{p}{p-l})\|u_1 - u_2\|. \tag{A.13}$$

Therefore, $M(u; S)$ is $(2p^2 - pl)/(p-l)$-gradient Lipschitz continuous.

5. By Eq. (A.8),

$$\|\nabla_u M(u)\| = \|p(u - w(u))\| = \|\nabla_w h(w)\| \le L. \tag{A.14}$$

$\square$

## A.3   PROOF OF THM. 5.1

The training loss is a standard result of runing GD on smooth objective function.

We focus on the proof of generalization bounds. Thm. 5.1 is not obtained from the work of (Hardt et al., 2016). Notice that

$$M(u; S) = \min_{w \in W} \frac{1}{n} \sum_{z \in S} K(w, u; z) \ne \frac{1}{n} \sum_{z \in S} \min_{w \in W} K(w, u; z).$$

$\min_u M(u; S)$ is not a finite sum problem. The analysis in (Hardt et al., 2016) can only be applied to finite sum problems. Thm. 5.1 requires a different proof. In summary, there are two steps:

1. Build the recursion from $\|u_S^t - u_{S'}^t\|$ to $\|u_S^{t+1} - u_{S'}^{t+1}\|$;
2. Unwind the recursion.

The main challenge comes from the first step, since the problem is not in the form of finite sum. To this end, we develop a new error bound and a different decomposition. We first introduce the following error bound.

**Lemma A.1.** *In weakly-convex case, for neighbouring $S$ and $S'$, we have*

$$\|w(u; S) - w(u; S')\| \le 2L/(n(p - \ell)).$$

*Proof.* By the $(p - l)$-strongly convexity of $K(w, u; S)$, we have

$$
\begin{aligned}
&(p - l)\|w(u; S) - w(u; S')\| \\
\le\ & \|\nabla K(w(u; S), u; S) - \nabla K(w(u; S'), u; S)\| \\
\le\ & \|\nabla K(w(u; S), u; S) - \nabla K(w(u; S'), u; S')\| \\
& + \frac{1}{n}\|\nabla h(w(u; S'), z_i)\| + \frac{1}{n}\|\nabla h(w(u; S'), z_i')\| \\
=\ & \frac{1}{n}\|\nabla h(w(u; S'), z_i)\| + \frac{1}{n}\|\nabla h(w(u; S'), z_i')\| \\
\le\ & \frac{2L}{n},
\end{aligned}
$$

where the second inequality is due to the definition of $K(w, u; S)$, the third one is due to the first-order optimally condition, and the last inequality is because of the bounded gradient of $h(w; z)$. $\square$

Next, we move to the proof of Thm. 5.1.

**Step 1.**

$$
\begin{aligned}
& \|u_S^{t+1} - u_{S'}^{t+1}\| \\
= & \|u_S^t - u_{S'}^t - \alpha_t(\nabla M(u_S^t; S) - \nabla M(u_{S'}^t; S'))\| \\
\leq & \|u_S^t - u_{S'}^t - \alpha_t(\nabla M(u_S^t; S) + \nabla M(u_{S'}^t; S))\| + \alpha_t\|\nabla M(u_{S'}^t; S') - \nabla M(u_{S'}^t; S)\| \\
\leq & \|u_S^t - u_{S'}^t\| + \alpha^t\|\nabla M(u_{S'}^t; S') - \nabla M(u_{S'}^t; S)\| \\
= & \|u_S^t - u_{S'}^t\| + \alpha^t p\|u_{S'}^t - u_{S'}^t - w(u_{S'}^t, S) + w(u_{S'}^t, S')\| \\
\leq & \|u_S^t - u_{S'}^t\| + \frac{2L\alpha_t}{n},
\end{aligned}
$$

where the second inequality is due to the non-expansive property of convex function , the last inequality is due to Lemma A.1.

**Step 2.** Unwinding the recursion, we have

$$
\|u_S^T - u_{S'}^T\| \leq \frac{2L\sum_{t=1}^T \alpha_t}{n}.
$$

$\square$

## A.4 PROOF OF LEMMA 5.1

Lemma 5.1 can be obtained from classical strong-convex optimization results. Since

$$
\|\nabla_w K(w, u; z)\| = \|\nabla_w h(w; z) + p(w - u)\| \leq L + pD_W,
$$

$K(w, u; z)$ has bounded gradient $L_K = L + pD_W$. By (Nemirovski et al., 2009), running SGD on $K(w, u; S)$ with stepsize $c_s \leq 1/s(p - l)$ iccurs an optimization error in

$$
E\|w_N - w(u)\|^2 \leq \frac{C_1^2}{N},
$$

where $C_1 = (L + pD_W)/(p - l)$.

## A.5 PROOF OF THM. 5.2

*Proof.* Let $A_{t+1} = \frac{1}{2}\|u^{t+1} - u^*\|^2$ and $a_{t+1} = \frac{1}{2}\mathbb{E}\|u^{t+1} - u^*\|^2$.

$$
\begin{aligned}
A_{t+1} & = \frac{1}{2}\|u^{t+1} - u^*\|^2 \\
& \leq \frac{1}{2}\|u^t - \alpha_t\nabla_u K(w_N^t, u^t; S) - u^*\|^2 \\
& \leq A_t + \frac{1}{2}\alpha_t^2 L_K^2 - \alpha_t\langle\nabla_u K(w_N^t, u^t; S), u^t - u^*\rangle \\
& = A_t + \frac{1}{2}\alpha_t^2 L_K^2 - \alpha_t\langle\nabla_u M(u^t; S), u^t - u^*\rangle \\
& \quad + \alpha_t\langle\nabla_u M(u^t; S) - \nabla_u K(w_N^t, u^t; S), u^t - u^*\rangle.
\end{aligned}
$$

By taking the expectation on both sides and Rearranging the terms, we have

$$
\begin{aligned}
& \alpha_t\mathbb{E}[M(u^t) - M(u^*)] \\
\leq & a_t - a_{t+1} + \frac{1}{2}\alpha_t^2 L_K^2 + \alpha_t\mathbb{E}\langle\nabla_u M(u^t; S) - \nabla_u K(w_N^t, u^t; S), u^t - u^*\rangle \quad \text{(A.15)}
\end{aligned}
$$

Since

$$\mathbb{E}\langle \nabla_u M(u^t; S) - \nabla_u K(w_N^t, u^t; S), u^t - u^* \rangle$$
$$\leq \|\nabla_u M(u^t; S) - \nabla_u K(w_N^t, u^t; S)\| \mathbb{E}\|u^t - u^*\|$$
$$\leq \frac{p C_1 D_W}{\sqrt{N}},$$

Eq. (A.15) becomes

$$\alpha_t \mathbb{E}[M(u^t) - M(u^*)]$$
$$\leq a_t - a_{t+1} + \frac{1}{2} \alpha_t^2 L_K^2 + \frac{\alpha_t p C_1 D_W}{\sqrt{N}}.$$

Let $N \geq T$. Take the summation over $t$. We obtain that

$$\sum_{t=1}^{T} \alpha_t \mathbb{E}[M(u^t) - M(u^*)]$$
$$\leq a_0 + \frac{1}{2} \sum_{t=1}^{T} \alpha_t^2 L_K^2 + \frac{\sum_{t=1}^{T} \alpha_t p C_1 D_W}{\sqrt{T}}.$$

There exists $t \leq T$, such that

$$\mathbb{E}[M(u^t) - M(u^*)] \leq \frac{a_0 + \frac{1}{2} \sum_{t=1}^{T} \alpha_t^2 L_K^2 + \frac{\sum_{t=1}^{T} \alpha_t p C_1 D_W}{\sqrt{T}}}{\sum_{t=1}^{T} \alpha_t}.$$

Considering constant step $\alpha \leq 1/\sqrt{T}$, we have $\alpha \leq 1/T\alpha$ and $\alpha\sqrt{T} \leq 1$. Therefore,

$$\mathbb{E}[M(u^t) - M(u^*)] \leq \frac{2a_0 + T\alpha^2 L_K^2 + 2\alpha\sqrt{T} p C_1 D_W}{2T\alpha}$$
$$\leq \frac{\|u^0 - u^*\|^2 + L_K^2 + 2p C_1 D_W}{2T\alpha}$$
$$= \frac{C_2}{T\alpha}.$$

Since $M(u; S)$ and $R_S(w)$ have the same global solutions, we can use both of them to measure the optimization error. Above is the optimization error defined in $M(u; S)$. Below we provide the optimization error defined in $R_S(w)$.

$$\mathbb{E}[R_S(w(u^t)) - R_S(w^*)] \leq \mathbb{E}[M(u^t) - M(u^*)] \leq \frac{C_2}{T\alpha}.$$

Notice that the choices of algorithm output are slightly different. Therefore, we have

$$\mathcal{E}_{opt} \leq \frac{C_2}{T\alpha},$$

where $C_2 = \|u^0 - u^*\|^2/2 + p C_1 D_W + (L + p D_W)^2/2$. $\qquad \square$

## A.6 PROOF OF THM. 5.3

*Proof.* We decompose $\|u_S^{t+1} - u_{S'}^{t+1}\|$ as

$$\mathbb{E}\|u_S^{t+1} - u_{S'}^{t+1}\|$$
$$= \mathbb{E}\|u_S^t - \alpha_t \nabla_u K(w_{N,S}^t, u_S^t; S) - u_{S'}^t + \alpha_t \nabla_u K(w_{N,S'}^t, u_{S'}^t; S')\|$$
$$\leq \mathbb{E}\|u_S^t - \alpha_t \nabla_u M(u_S^t; S) - u_{S'}^t + \alpha_t \nabla_u M(u_{S'}^t; S')\|$$
$$+ 2\alpha_t \mathbb{E}\|\nabla_u K(w_{N,S}^t, u_S^t; S) - \nabla_u M(u_S^t; S)\|$$
$$\leq \mathbb{E}\|u_S^t - u_{S'}^t\| + \frac{2L\alpha_t}{n} + 2\alpha_t p \mathbb{E}\|w_N^t - w(u^t)\|$$
$$\leq \mathbb{E}\|u_S^t - u_{S'}^t\| + \frac{2L\alpha_t}{n} + 2\alpha_t p \frac{C_1}{\sqrt{N}}.$$

Let $N \geq n^2$. Unwind the recursion and let $u^T$ be the output of the algorithm. We have

$$
\begin{aligned}
\mathcal{E}_{gen} &\leq L\mathbb{E}\|u_S^T - u_{S'}^T\| \\
&\leq \frac{L(2L + 2C_1 p) \sum_{t=1}^T \alpha_t}{n} \\
&= \frac{C_3 \sum_{t=1}^T \alpha_t}{n}.
\end{aligned}
$$

If we choose $w(u^T)$ to be the algorithm output, we have

$$
\begin{aligned}
\mathcal{E}_{gen} &\leq L\mathbb{E}\|w(u_S^T; S) - w(u_{S'}^T; S')\| \\
&= L\mathbb{E}\|u_S^T - \frac{1}{p}\nabla M(u_S^T, S) - u_{S'}^T - \frac{1}{p}\nabla M(u_{S'}^T; S'))\| \\
&\leq L\mathbb{E}\|u_S^T - u_{S'}^T\| + \frac{2L^2}{np} \\
&\leq \frac{L(2L + 2C_1 p)\sum_{t=1}^T \alpha_t}{n} + \frac{2L^2}{np} \\
&= \mathcal{O}\left(\frac{\sum_{t=1}^T \alpha_t}{n}\right). \tag{A.16}
\end{aligned}
$$

where the first equality is due to $\nabla M(u; S) = p(u - w(u))$, the second inequality is due to the non-expansive propertiy of $M(u; S)$. $\qquad\square$

## B  WEAKLY-CONVEX CASES

Our main result can be extended to weakly convex cases.

**Theorem B.1.** *Assume $h$ is a weakly-convex, L-Lipschitz function. Suppose we run GD on the smoothed surrogate adversarial loss $M(u; S)$ defined in Eq. (5.2) with diminishing stepsize $\alpha \leq (p - l)/(2p^2 - pl)t$ for $T$ steps. Then, the generalization gap satisfies*

$$
\mathcal{E}_{gen} \leq \mathcal{O}\left(\frac{2L^2 T}{(2p - l)n}\right). \tag{B.1}
$$

In this case, the bound also does not contain a non-vanishing term. The proof based on the error bound (Lemma A.1) and the decomposition in the proof of Thm. 5.1.

Proof:

**Step 1.**

$$
\begin{aligned}
&\|u_S^{t+1} - u_{S'}^{t+1}\| \\
=\ &\|u_S^t - u_{S'}^t - \alpha_t(\nabla M(u_S^t; S) - \nabla M(u_{S'}^t; S'))\| \\
\leq\ &\|u_S^t - u_{S'}^t - \alpha_t(\nabla M(u_S^t; S) + \nabla M(u_{S'}^t; S))\| + \alpha_t\|\nabla M(u_{S'}^t; S') - \nabla M(u_{S'}^t; S)\| \\
\leq\ &\|u_S^t - u_{S'}^t\| + \alpha_t\|\nabla M(u_S^t; S) - \nabla M(u_{S'}^t; S)\| + \alpha^t\|\nabla M(u_{S'}^t; S') - \nabla M(u_{S'}^t; S)\| \\
\leq\ &(1 + \alpha_t\beta)\|u_S^t - u_{S'}^t\| + \alpha^t\|\nabla M(u_{S'}^t; S') - \nabla M(u_{S'}^t; S)\|, \tag{B.2}
\end{aligned}
$$

where the first and second inequalities are due to triangular inequality. The last inequality is due to the gradient Lipschitz of $M(u; S)$ and $\beta = (2p^2 - pl)/(p - l)$. Then,

$$
\begin{aligned}
&\alpha^t\|\nabla M(u_{S'}^t; S') - \nabla M(u_{S'}^t; S)\| \\
=\ &\alpha^t p\|u_{S'}^t - u_{S'}^t - w(u_{S'}^t, S) + w(u_{S'}^t, S')\| \\
\leq\ &\frac{2Lp\alpha_t}{(p - l)n}, \tag{B.3}
\end{aligned}
$$

where the first inequality is due to the form of $\nabla M(u; S)$, the last equality is due to Lemma A.1.

Combining Eq. (B.2) and (B.3), we have

$$
\begin{aligned}
&\|u_S^{t+1} - u_{S'}^{t+1}\| \\
\leq\quad &(1 + \alpha_t \beta)\|u_S^t - u_{S'}^t\| + \frac{2Lp\alpha_t}{(p-l)n}.
\end{aligned}
\tag{B.4}
$$

**Step 2.**  Let $\alpha_t \leq (p-l)/(2p^2 - pl)t$,

$$
\begin{aligned}
&\|u_S^{t+1} - u_{S'}^{t+1}\| \\
\leq\quad &\exp(1/t)\|u_S^t - u_{S'}^t\| + \frac{2L}{(2p-l)n}.
\end{aligned}
\tag{B.5}
$$

Then, following the proof, for example, of Xiao et al. (2022), we have

$$
\mathcal{E}_{gen} \leq \mathcal{O}\left(\frac{2L^2 T}{(2p-l)n}\right).
$$

$\square$

