# OpenReview forum: "Smoothed-SGDmax: A Stability-Inspired Algorithm to Improve Adversarial Generalization"
_ICLR.cc/2023/Conference — Submitted to ICLR 2023_

### Official Review · Reviewer_Kcyd · 2022-10-21

**Confidence:** 3
**Correctness:** 4
**Technical Novelty And Significance:** 3
**Empirical Novelty And Significance:** 3
**Recommendation:** 6

**Clarity, Quality, Novelty And Reproducibility:**

Overall the paper is well and clearly written.  The results are also novel and original.

There may still be some room for improving the presentation and bringing out more insight. For example, how the non-smoothness of the adversarial loss impacts the algorithmic stability of SGDMax is not discussed. As this appears a key motivation justifying the proposal of Smoothed-SGDMax, its deserves some careful discussion in my opinion.





**Strength And Weaknesses:**

**Strength**

* Despite robust overfitting has been observed for quite some time and efforts have been spent to understand its root cause, the problem remain mostly open. This paper is a new effort towards solving this important problem, namely, understanding the generalization behaviour of the SGDMax algorithm in adversarial training.

* The paper provides novel perspectives (a lower bound) in characterizing the generalization gap with respect to the optimization gap. The proposed Smoothed-SGD algorithm is intuitively sensible, theoretically justified and empirically validated.

* The theoretical analysis appears solid.

** Weakness **

* Requiring the functions in class ${\cal H}$ to be convex significantly limits the applicability of the developed results.
* Comparing with SGDMax, Smoothed-SGDMax does show improved generalization. However even with Smoothed-SGDMax, there still appears a significant gap between training and test robust accuracies (e.g., green and red curves in Figure 1). Such a gap appears much larger than its counter-part in (regular) training.  Then it seems that non-smoothness can not fully explain robust overfitting.


**Summary Of The Paper:**

The paper studies the generalization problem in adversarial training. Specifically, it presents a minimax lower bound of the generalization gap, proposes Smoothed-SGDMax that achieves the bound. Connection to stochastic weight averaging (SWA) is also presented.

**Summary Of The Review:**

Overall the paper is mostly well written with novel results towards better understanding of an important problem (robust overfitting) in deep learning, although the results are built on strong assumptions, rather distant from practice.

---

> ### Author Response · Authors · 2022-11-14
> **Responds to Reviewer Kcyd**
>
> We thank Reviewer Kcyd for the comments and question. Below we answer your questions.
>
> ___
>
> **Q1:** Requiring the functions in class H to be convex significantly limits the applicability of the developed results.
>
> A: Thanks for the question, Firstly, we extend the result to weakly-convex cases. The generalization bound is
>
> $$\mathcal{E}_{gen}\leq
> \frac{2L^2 T}{(2p-l)n},$$
>
> which still does not contain the non-vanishing term. We are still proofreading the proof and will present it in the next updated version.
>
> Secondly, from the work of Hardt et al., 2016, it is shown that stability analysis on convex and non-convex settings provide similar interpretations on generalization. Therefore, stability analysis on convex settings could provide some insights for empirical machine learning.
>
> ___
> **Q2:** There still appears to be a significant gap between training and test robust accuracies ... Then it seems that non-smoothness can not fully explain robust overfitting.
>
> A: Thanks for the question. We agree that non-smoothness can not fully explain robust overfitting. The gap comes from the assumption of H and the properties of DNNs in practice. Actually, the large adversarially robust generalization gap has been a key problem in adversarial learning in recent years. Uncovering the reasons is still an open problem. Our work provides an analysis from one perspective.
>
> ___
> **Q3:** There may still be some room for improving the presentation and bringing out more insight.
>
> A: Thanks for the suggestion. We tried to convey our idea clearer in the updated version. How the non-smoothness of the adversarial loss impacts the algorithmic stability of SGDMax is discussed in previous studies (e.g., Xing et al., 2021) , we rewrite the subsection 3.1 to present the previous studies.

---

> > ### Comment · Reviewer_Kcyd · 2022-12-13
> > **Thanks for the response**
> >
> > I have read the authors' response and found it overall reasonable, particularly regarding convexity of the function class.
> >
> > I would like to keep my score.

---

### Official Review · Reviewer_rzkp · 2022-10-24

**Confidence:** 3
**Correctness:** 2
**Technical Novelty And Significance:** 2
**Empirical Novelty And Significance:** 3
**Recommendation:** 5

**Clarity, Quality, Novelty And Reproducibility:**

I'll give comments on each topic.

(Minimax lower bound)
First, the statement of Theorem 4.1 looks weird. The authors take maximization over h inner the minimization over the algorithms. It implicitly assumes that the learner does not know the adversarial loss h, which conflicts with the setting in which the learner knows h. In Chen et al.'s analyses, they take the inner maximization over the distribution D,  which is unknown to the learner. Even if the statement is fixed so that the maximization is over D, I cannot find any difference from Chen et al., 2018. It seems that the originality of Theorem 4.1 is significantly low.

Second, and more importantly, the lower bound in Theorem 4.1 might be incorrect due to the possibly invalid assumption of Eq. 4.1. The results of Xing et al. and Xiao et al. guarantee an adversarial loss for convex and Lipschitz losses is an element of H in Eq. 4.1. It is not proved that there is a set of g such that the induced adversarial losses are equivalent to H in 4.1. We might need to think of a smaller set of losses as the adversarial losses, by which the minimax error might be smaller. Theorem 4.1 may be correct under the assumption of Eq. 4.1, while it might not be the minimax lower bound on the adversarial loss minimization.

(Exact algorithm)
I cannot find any difference from Lemma 2.5 in the following paper:
- D. Davis and D. Drusvyatskiy. Proximal methods avoid active strict saddles of weakly convex functions. Foundations of Computational Mathematics, 2022.
It is better to clarify the difference between Lemma 5.1 and the result of Davis and Drusvyatskiy.

Given Lemma 5.1, Theorem 5.1 is just an application of Hardt et al.'s result. I cannot find any novelty in this analysis.

(Inexact algorithm)
The inexact version of the algorithm and its analyses may be novel. It is nice that the inexact one achieves the same rate as the exact one.

(Experimental results)
The experimental results of Figure 2 adequately demonstrate the benefit of the proposed algorithm. One thing unclear is why the robust training accuracy of AT, whereas the present algorithm achieves high accuracy. Could the authors clarify this point?

I have one thing unclear in Figure 3. What does 10:0 mean? Does it mean no training data?

Minor comments:
- The authors claim that SGDmax does not achieve the minimax lower bound with the evidence of Theorem 4.2. However, Theorem 4.2 just demonstrates the upper bound, which might have a tightness issue. Fortunately, the original work reveals the matching algorithm-specific lower bound, which is better to support the claim.

**Strength And Weaknesses:**

Strength:
- The minimax optimality in the adversarial loss minimization is crucial for understanding adversarial robustness.
- The inexact version of the proposed algorithm is novel.
Weakness:
- The minimax lower bound result has issues in its originality and correctness.
- The exact version of their algorithm has an originality issue.

**Summary Of The Paper:**

The authors investigate the generalization error in optimizing the expected adversarial loss. They utilize the generalization-optimization decomposition and techniques from stability analyses to reveal the minimax optimal optimization algorithm for the adversarial loss. For the lower bound, they show that any algorithm with the convergence rate $s(T)$ suffers from the generalization gap at least $s(T)/n$, where n is the sample size. For the upper bound, they develop an optimization algorithm employing the Moreau envelope to smoothen the adversarial loss function so that the SGD is stabilized. They show that it achieves the optimal generalization gap with $s(T)=T$. Also, they analyzed the optimization algorithm with the approximated Moreau envelope, which gives the same optimal rate. The empirical evaluations demonstrate that their algorithm can avoid robust overfitting.


**Summary Of The Review:**

The minimax optimality of the adversarial loss minimization is a well-motivated and interesting direction. However, I found some issues or low originality in most results. Also, while their main idea is to employ the Moreau envelope, it is not original, as Hardt et al. discuss it as a way to control stability. I thus recommend the rejection.

---

> ### Author Response · Authors · 2022-11-14
> **Response to Reviewer rzkp (2/2)**
>
> **Q5:** (Exact algorithm) I cannot find any difference between Lemma 5.1 and the Lemma 2.5 of Davis and Drusvyatskiy 2022.
>
> A: Firstly, Lemma 5.1 are properties or their direct results of Moreau envelopes. We add `The proof of Lemma 5.1 is due to Rockafellar (1976) and also provided in Appendix A.2' below Lemma 5.1 in the updated version.
>
>
> Secondly, as suggested by reviewer rzkp, we carefully read the paper of Davis and Drusvyatskiy 2022. They also mentioned that the proof of Lemma 2.5 follows, for example, from R.A.Poliquinand and R.T.Rockafellar, 1996. Therefore, it is not surprising that we both use the properties of Moreau envelopes.
> ___
> **Q6:** One thing unclear is why the robust training accuracy of AT, whereas the present algorithm achieves high accuracy. Could the authors clarify this point?
>
>
> A: In our understanding, the question is why the present algorithm achieves higher accuracy than the baseline algorithm? If it is not the question, please correct us.
>
> In the experiment on CIFAR-100 dataset, the algorithm does achieve higher training accuracy. However, in the experiments on CIFAR-10 (Fig. 1), the training accuracy of Smoothed-SGDmax is lower. In the experiment on SVHN (Fig. 2(a)), the training accuracy of SGDmax and Smoothed-SGDmax is similar. Therefore, we cannot make any conclusion on training accuracy in practice.
>
>  ___
> **Q7:** I have one thing unclear in Figure 3. What does 10:0 mean? Does it mean no training data?
>
> A: It is labeled data to unlabeled data. We fixed it in the updated version.
> ___
>
> **Q8:** Theorem 4.2 just demonstrates the upper bound.
>
> A: Thanks for the question. We rewrite this subsection. Now, the lower bounds and upper bounds in previous studies are all presented in Table 2.

---

> ### Author Response · Authors · 2022-11-14
> **Response to Reviewer rzkp (1/2)**
>
> We thank Reviewer rzkp for the comments and questions.  We first answer the most critical comments and questions. After that, we answer other questions one by one.
> ___
> **Comment 1:** Given Lemma 5.1, Theorem 5.1 is just an application of Hardt et al.'s result. I cannot find any novelty in this analysis.
>
> A: Thanks for the comment. However, Theorem 5.1 is not an application of Hardt et al.,'s result. The analysis is new, and the proof is different. It is our first main result.
>
> In summary, Hardt et al. 2016 considered SGD (and other stochastic algorithms) on finite sum problems,
>
> $$\frac{1}{n}\sum f(w,z_i),$$
>
> where $f$ is smooth.
>
> In Theorem 5.1, we consider GD on $M(u;S)$, which is not in the form of a finite sum.
>
>
> $$M(u;S)=min_w[\frac{1}{n}\sum h(w,z_i)+\frac{p}{2}||w-u||^2]\neq \frac{1}{n}\sum min_w[ h(w,z_i)+\frac{p}{2}||w-u||^2]. $$
>
> We refer to running SGD on the last expression as Alg. 2. Hardt et al. 2016's analysis can only be applied to Alg.2. However, Alg.2 is not guaranteed to converge. In words, it is something like "trains n DNNs on each sample and takes the average". It is also not implementable in practice.
>
> To obtain a stability-based generalization bound of running GD on $M(u;S)$, we provide new error bounds and different analysis. Details are provided in Appendix A.3.
> ___
> **Comment 2:** their main idea is to employ the Moreau envelope, it is not original, as Hardt et al. discuss it as a way to control stability.
>
> A: Thanks for the comment. It is not discussed in Hardt et al. 2016. The work of Hardt et al. 2016 discussed a proximal point algorithm that used the Moreau envelopes. However, both the algorithm and the analytical tools are different. More importantly, the proximal point algorithm is not guaranteed to eliminate the generalization error floor.
>
> The proximal update is to apply an update rule,
> $$P_{f,\alpha}=argmin_u||w-u||^2+\alpha f(u),$$
> after a stochastic gradient update. A stability analysis of Proximal update is given
> in Hardt et al. (2016), Def. 4.5 and Lemma 4.6. It is proved that the proximal update is 1-expansive
> if f is convex. Therefore, the generalization bound of the proximal update is no larger than that of
> SGD.
>
> If we apply this analysis to non-smooth cases, the generalization bound of proximal update is no larger than that of SGD, which contains an error floor. Therefore, the proximal update is not guaranteed to eliminate the error floor. We add the discussion of proximal update in section 5.4 in the updated version.
>
> In summary, the work of Hardt et al. 2016 shows that using Moreau envelopes is not always guaranteed
> to improve generalization. It helps justify that our results are nontrivial.
> ___
> We answer other questions one by one.
>
> **Q1:** (Minimax lower bound) First, the statement of Theorem 4.1 looks weird.
>
> A: Thanks for the question. First of all, max_h is a typo. It should be max_D. Secondly, Theorem 4.1 is not the main result of our paper. It is used to show the tightness of our generalization bounds, which are our main results. To avoid misleading, we use Proposition 4.1 in the updated version.
> ___
> **Q2:** It is not proved that there is a set of g such that the induced adversarial losses are equivalent to H in 4.1.
>
> A: Thanks for the question. We agree that the class of adversarial loss is a subset of non-smooth loss. However, it is hard to characterize the smoothness of the adversarial loss function precisely. The work of Xing et al., 2021 and Xiao et al., 2022 used different smoothness assumptions. In our work, the general non-smooth assumption follows the work of Xing et al., 2021.
>
> ___
> **Q3:** We might need to think of a smaller set of losses as the adversarial losses, by which the minimax error might be smaller.
>
> A: For smooth function, the minimax lower bound has the same order in Eq. 4.2 (Nemirovskij & Yudin, 1983). Since adversarial loss is not smooth, we believe that the minimax error might not be smaller.
> ___
> **Q4:** Theorem 4.1 may be correct under the assumption of Eq. 4.1, while it might not be the minimax lower bound on the adversarial loss minimization.
>
> A: Thanks for the question. We will fix the mathematical description. We use minimax lower bound on non-smooth loss minimization in the updated version.

---

> ### Author Response · Authors · 2022-12-07
> **Open to discussion**
>
> Dear Review rzkp,
>
> Thanks again for the comments and questions. The deadline of discussion Stage 2 is coming soon. Since we found there are some misunderstandings on the contribution/novelty, we would appreciate it if you could engage in the discussion to tell whether we have addressed your questions and/or you have any other questions.
>
> Authors

---

### Official Review · Reviewer_9aTN · 2022-10-24

**Confidence:** 3
**Correctness:** 3
**Technical Novelty And Significance:** 2
**Empirical Novelty And Significance:** 2
**Recommendation:** 5

**Clarity, Quality, Novelty And Reproducibility:**

The paper is clearly not well-written.  I recommend that the authors carefully
proof-read the manuscript and fix the various typos and issues in notation.

**Strength And Weaknesses:**

Strengths

1. The paper studies the important and well-motivated problem of adversarial
training and proposes a new method that improves generalization.

Weaknesses

1. The paper is hard to parse and follow and there are numerous typos and
inconsistencies in the notation.  One of the main results of this work (Theorem
4.1) seems to be a direct corollary of prior work but still the proof provided
in Appendix A.1 is hard to follow.

2. I did not find the Smoothed-SGDMax method proposed by the authors
particularly novel since approximating non-smooth terms with their
corresponding Moreau envelopes is pretty standard.


Typos and other Issues

Theorem 4.2 does not show that SGDmax does not achieve the minimax lower bound.
It only gives an upper bound for the generalization error of SGD that does
not match the lower bound of Theorem 4.1.

Page 2, next to Figure 1. "By contrast, Our approach"

Page 3, First paragraph of Related Work. "This has led to a series of work aimed at training..."

Page 13, Proof of Theorem 4.1. "Where $C_4$ is a universal constant". There is no constant $C_4$ in the expression (A.2).

Page 13, Proof of Lemma 5.1. The sentence in Element 1 of the proof does not parse.

Page 13, "Then, take the derivative of $M(u)$ with respect to $u$, we have"



**Summary Of The Paper:**

This work studies the effect of using adversarially robust on generalization
error.  In particular they show that the generalization error achieved by
SGDmax contains a constant term that does not tend to 0 as the sample size
increases.  They show that they can remove this constant term in the error by
using a smooth variant of SGDmax.



**Summary Of The Review:**

Overall, I found the paper not well-written and I believe that it would benefit
a lot by careful proofreading. I do not think that in its current state this
work is ready for publication.   Perhaps after a revision, the contributions of
this would also become clearer.  Given the fact that the paper is not
well-written and the contributions do not seem very exciting, I am currently
inclined towards rejection but I am willing to reconsider if the authors and/or
other reviewers can prove otherwise.

---

> ### Author Response · Authors · 2022-11-14
> **Response to Reviewer 9aTN**
>
> We thank Reviewer 9aTN for the comments and questions. Below we answer the questions.
> ___
> **Q1:** The paper is hard to parse and follow, and there are numerous typos and inconsistencies in the notation.
>
> A: Thanks for the comments. We tried our best to fix the typos and to make our idea clearer in the updated version. We will keep proofreading the paper.
> ___
> **Q2:** One of the main results of this work (Theorem 4.1) seems to be a direct corollary of prior work, but still, the proof provided in Appendix A.1 is hard to follow.
>
> A: Thanks for the question. First of all, Theorem 4.1 is not the main result of our paper. It is used to show the tightness of our generalization bounds, which are our main results. To avoid misleading, we use Proposition 4.1 in the updated version. Also, we rewrite the proof in Appendix A.1. We hope it is clearer.
> ___
> **Q3:** I did not find the Smoothed-SGDMax method proposed by the authors particularly novel since approximating non-smooth terms with their corresponding Moreau envelopes is pretty standard.
>
> A: Thanks for the question. Moreau envelope function is indeed a common tool used in optimization for many years, but it is not connected to improving generalization in the previous studies to our knowledge.
>
> Our main contribution and novelty are to prove that the non-vanishing term (which is also referred to as error floor in the updated version) can be eliminated by properly designed algorithms.
>
>
> Using Moreau envelopes to solve this generalization problem contains several non-trivial steps. We try to make the idea clearer.
>
> **1 Even though the non-vanishing term comes from non-smoothness, it does not imply that smoothed algorithms can solve this problem.**
>
> Stability analysis on some smoothed algorithms has been
> studied recently. It includes noise-SGD and differential privacy-SGD (Bassily et al. (2020)), adding
> noise to weight and data (Xing et al. (2021b)), stochastic weight averaging, and cyclic learning rate
> (Xiao et al. (2022)). Unfortunately, they found that these smoothed algorithms cannot eliminate the
> generalization error floor.
>
> **2 Given the hint using the Moreau envelope, it is not obvious how to use it to solve the generalization problem.**
>
> We consider two more algorithms.
>
> The first one is the proximal point method (a popular algorithm using the Moreau envelope). We called it Alg. 1.
>
> The proximal update is to apply an update rule,
> $$P_{f,\alpha}=argmin_u||w-u||^2+\alpha f(u),$$
> after a stochastic gradient update. Both proximal update and Smoothed-SGDmax use the Moreau
> envelope function, but the algorithms are different. A stability analysis of Proximal update is given
> in Hardt et al. (2016), Def. 4.5 and Lemma 4.6. It is proved that the proximal update is 1-expansive
> if f is convex. Therefore, the generalization bound of the proximal update is no larger than that of
> SGD. In non-smooth cases, SGD incurs an error floor. The proximal update is not guaranteed to be able
> to eliminate the error floor.
>
> For the second one, let's recap the definition of $M(u;S)$,
>
> $$M(u;S)=min_w[\frac{1}{n}\sum h(w,z_i)+\frac{p}{2}||w-u||^2]\neq \frac{1}{n}\sum min_w[ h(w,z_i)+\frac{p}{2}||w-u||^2]. $$
>
> We refer to running SGD on the last expression as Alg. 2. Using Hardt et al. 2016's result, it is easy to obtain the generalization bound of Alg.2. However, Alg.2 is not guaranteed to converge. In words, it is something like "trains n DNNs on each sample and takes the average". It is also not implementable in practice.
>
>
> Therefore, naively using Moreau envelopes may have generalization issues (Alg. 1) or convergence issues (Alg. 2).
>
> To address both issues, we consider running SGD on $M(u;S)$. Then, no existing results can be applied. It is also not obvious whether Smoothed-SGDmax can solve the generalization problem.
>
>
> **3 Even given the algorithm Smoothed-SGDmax, proving that it can eliminate the generalization error floor is non-trivial.**
>
> To obtain a stability-based generalization bound of running GD on $M(u;S)$, we provide a different analysis. Details are provided in Appendix A.3. The proof is not adopted from previous studies (e.g., Hardt et al., 2016). To obtain the optimal generalization bounds, we develop new error bounds, and new analysis in the proof of Thm. 5.1 and 5.3.
>
> Overall, from the description, we guess you might think using the Moreau envelope function to smooth the loss (Step 1) and improving the generalization bound (step 2) are just two trivial steps. If our guesses are wrong, please correct us. However, none of them are trivial. We tried to make the idea clearer in the updated version. And we hope our explanation can address your questions.
> ___
> **Q4:** Theorem 4.2 does not show that SGDmax does not achieve the minimax lower bound.
>
> A: Thanks for the question. We rewrite this subsection. Now, the lower bounds and upper bounds in previous studies are all presented in Table 2.

---

> ### Author Response · Authors · 2022-12-07
> **Open to discussion**
>
> Dear Review 9aTN,
>
> Thanks again for the comments and questions. The deadline of discussion Stage 2 is coming soon. Since we found there are some misunderstandings on the contribution/novelty, we would appreciate it if you could engage in the discussion to tell whether we have addressed your questions and/or you have any other questions.
>
> Authors

---

### Official Review · Reviewer_xxrH · 2022-10-25

**Confidence:** 3
**Correctness:** 4
**Technical Novelty And Significance:** 3
**Empirical Novelty And Significance:** 2
**Recommendation:** 6

**Clarity, Quality, Novelty And Reproducibility:**

I think the paper is well-written and easy to follow. The results concern both lower and upper bounds, which reveals both novelty and significance.

**Strength And Weaknesses:**

Strength:
1. The paper is well-written and the flow is clear
2. The proposed algorithm successfully overcomes the nonvanishing generalization errors in existing SGDmax results, while still share the same convergence guarantee

Weakness:
1. The tools in algorithm building are not that novel.
2. The problem setting is the convex regime, which is a bit restricted (to be a bit picky)

**Summary Of The Paper:**

This paper studied the generalization performance of adversarial training from the stability perspective. They first derived the minimax lower bound of generalization gap for convex Lipschitz functions, which identifies the gap compared to existing generalization results of SGDmax; then authors proposed Smoothed-SGDmax which shares the same convergence guarantee while achieves the aforementioned (vanishing) minimax lower bound, i.e., fix the gap.

**Summary Of The Review:**

Generally I don't have immediate questions to ask.

Some typos:
1. Theorem 4.2, $\alpha_t$ and $\alpha$, do you suggest that you are always using constant stepsize $\alpha_t\equiv\alpha$?

---

> ### Author Response · Authors · 2022-11-14
> **Response to Reviewer xxrH**
>
> We thank Reviewer xxrH for the comments and questions. Below we answer your questions.
> ___
> **Q1:** The tools in algorithm building are not that novel.
>
> A: Thanks for the question. Moreau envelope function is indeed a common tool used in optimization for many years but it is not connected to generalization in the previous studies.
>
> Our main contribution and novelty is to prove that the non-vanishing term (which is also referred to error floor in the updated version) can be eliminate by proper designed algorithms. Using Moreau envelopes to solve this generalization problem contain several non-trivial steps.
>
> 1) 1 Even though the non-vanishing term comes from non-smoothness, it does not imply that smoothed algorithms can solve this problem.
>
> 2)  Given the hint using the Moreau envelope, it is not obvious how to use it to solve the generalization problem.
>
> 3) Even given the algorithm Smoothed-SGDmax, proving that it can eliminate the generalization error floor is non-trivial.
>
> The question is related to Q3 from Reviewer 9aTN. We refer to the detail explanation of this three points there.
>
> We hope our explanation can make the idea clearer.
> ___
> **Q2:** The problem setting is the convex regime, which is a bit restricted (to be a bit picky)
>
> A: Thanks for the question, Firstly, we extend the result to weakly-convex cases. The generalization bound is
>
> $$\mathcal{E}_{gen}\leq
> \frac{2L^2 T}{(2p-l)n},$$
>
> which still does not contain the non-vanishing term. We are still proofreading the proof and will present it in the next updated version.
>
> Secondly, from the work of Hardt et al., 2016, it is shown that stability analysis on convex and non-convex settings provide similar interpretation on generalization. Therefore, stability analysis on convex settings could provide some insight for empirical machine learning.
> ___
> **Q3:** do you suggest that you are always using constant stepsize？
>
> A: Thanks for the question. First of all, we fixed the typos in the update version. We mainly use fixed step size in the theoretical results. Secondly, in convex case, whether there is $T\alpha$ (fixed step size) or $\sum \alpha_t$ (varying step size) in the generalization bound does not affect the analysis. Therefore, we use fixed step size in the updated version.

---

> > ### Comment · Reviewer_xxrH · 2022-12-13
> > **Thank you**
> >
> > Thank you for the response which clarifies my confusion, I will keep my score.

---

### Author Response · Authors · 2022-11-18
**Summary of revision**

Summary of revision:

Section 1:

1) We rewrite the main question of the paper.

2) We add a paragraph to discuss the previous analysis of smoothed algorithms.

3) Main contribution: our main contribution is to show that the error floor can be eliminated by properly designed algorithms.

Section 2: We rewrite the paragraph on learning theory and uniform stability.

Section 3: We present the previous studies on generalization error floors, including both upper bounds and Lower bounds, in Sec. 3.1.

Section 4:

1) Def. 4.1 is called training loss in the updated version for consistent terminology. We fix the typos in the minimax lower bound.

2) Thm. 4.1 is now presented in Prop. 4.1 to avoid misleading.

Section 5.1:

1) We add 'The proof of Lemma 5.1 is due to (Rockafellar, 1976) and also provided in Appendix A.1.'

2) We add more discussion on Thm. 5.1. It is not an application of Hardt et al.,' results since $M(u;S)$ is not a finite sum problem.

3) The analysis is not adopted from previous studies. We provide the idea of the proof in the main text.

Section 5.4:

1) We add the comparison to the proximal update, which is discussed in Hardt et al., 2016. In this analysis, using the Moreau envelope function is not guaranteed to eliminate the generalization floor.


Section 6: We fix the typo 'Labeled to unlabeled data.'

Appendix A: We polish the proof to make our idea clear.

Appendix B: We add the extension to weakly-convex cases.

Overall, we hope our revised version conveys our idea more clearly.

Most importantly, we hope it is clear that

1) The use of Moreau envelopes is nontrivial, naively using Moreau envelopes incurs either optimization or generalization issues.

2) The proof is novel. It is not from previous studies.

We hope reviewers can engage in the discussion to see whether we have addressed the questions.

---

### Decision · Program_Chairs · 2023-01-20

**Decision:**

Reject

**Justification For Why Not Higher Score:**

technical novelty is lacking. The results seem not hard to achieve.

**Justification For Why Not Lower Score:**

N/A

**Metareview: Summary, Strengths And Weaknesses:**

This paper studied the generalization performance of adversarial training from the stability perspective. The authors first derived a minimax lower bound of generalization gap, and then proposed a smoothed-SGDmax algorithm using the technique of Moreau envelop which enjoys a generalization bound that matches the lower bound. In particular, smoothed-SGDmax algorithm eliminates the "the generalization error floor" of standard SGDmax algorithm without the smoothing technique proposed in this paper.

During the AC-reviewer meeting, all reviewers agree on that fact this paper studies an important problem is important. The results are also interesting, reviewers learned something new from the paper that potentially worth publishing. However, after seeing the updated version of the draft, reviewers still believe there remain a fair amount of mathematical typos in paper which hurts the quality. (Despite reviewers agree that the idea of Moreau envelope may not have been extensively used in the problem considered in this paper before, the proof is different from Hardt et al., 2016) reviewers still feel the technical novelty is a bit lacking, and it seems the results follow from the relatively standard analysis even though it is not exact the same from the existing papers. It's the combination of these two make this paper borderline leaning towards rejection.

I will recommend the authors to (1) fix the typos (I can also easily see a few even in the revised version); (2) highlight the technical challenges answering "why the results actually take a lot effort to get" and then maybe retry a similar venue.

**Summary Of Ac-Reviewer Meeting:**

Strength:
1. (40%). This paper provides new results to this problem, which makes contribution to the field.

Weakness:
1. (10%) Writing (still not good in the second version), mathematical typos.
2. (50%) Idea is simple, is not novel enough (especially the Moreau envelope part)